# Local Delivery of Gemcitabine Inhibits Pancreatic and Cholangiocarcinoma Tumor Growth by Promoting Epidermal Growth Factor Receptor Degradation

**DOI:** 10.3390/ijms21051605

**Published:** 2020-02-26

**Authors:** Sung Ill Jang, Sungsoon Fang, Yi-Yong Baek, Don Haeng Lee, Kun Na, Su Yeon Lee, Dong Ki Lee

**Affiliations:** 1Department of Internal Medicine, Gangnam Severance Hospital, Yonsei University College of Medicine, Seoul 06273, Korea; AEROJSI@yuhs.ac (S.I.J.); YYBAEK@yuhs.ac (Y.-Y.B.); LSY83@yuhs.ac (S.Y.L.); 2Severance Biomedical Science Institute, BK21 Plus Project for Medical Science, Gangnam Severance Hospital, Yonsei University College of Medicine, Seoul 06273, Korea; sfang@yuhs.ac; 3Department of Internal Medicine, Inha University Hospital, Inha University College of Medicine, Incheon 22212, Korea; ldh@inha.ac.kr; 4Utah-Inha DDS & Advanced Therapeutics Research Center, Incheon 22212, Korea; 5Department of Biotechnology, The Catholic University of Korea, Bucheon-si 14662, Korea; kna6997@catholic.ac.kr

**Keywords:** biliary duct carcinoma, pancreatic cancer, drug-eluting stent, gemcitabine, EGFR, angiogenesis

## Abstract

Gemcitabine is clinically used to treat certain types of cancers, including pancreatic and biliary cancer. We investigated the signal transduction pathways underlying the local antitumor effects of gemcitabine-eluting membranes (GEMs) implanted in pancreatic/biliary tumor-bearing nude mice. Here, we report that GEMs increased the E3 ubiquitin ligase c-CBL protein level, leading to degradation of epidermal growth factor receptor (EGFR) in SCK and PANC-1 cells. GEMs decreased the RAS and PI3K protein levels, leading to a reduction in the protein levels of active forms of downstream signaling molecules, including PDK, AKT, and GSK3β. GEM reduced proliferation of cancer cells by upregulating cell cycle arrest proteins, particularly p53 and p21, and downregulating cyclin D1 and cyclin B. Moreover, GEMs reduced the levels of proangiogenic factors, including VEGF, VEGFR2, CD31, and HIF-1α, and inhibited tumor cell migration and invasion by inducing the expression of E-cadherin and reducing that of N-cadherin, snail, and vimentin. We demonstrated that local delivery of gemcitabine using GEM implants inhibited tumor cell growth by promoting c-CBL-mediated degradation of EGFR and inhibiting the proliferation, angiogenesis, and epithelial–mesenchymal transition of pancreatic/biliary tumors. Use of gemcitabine-eluting stents can improve stent patency by inhibiting the ingrowth of malignant biliary obstructions.

## 1. Introduction

Patients with unresectable malignant biliary obstruction require biliary drainage to improve survival time and quality of life [1,2]. Use of endoscopically implanted self-expandable metal stents (SEMSs) is the treatment of choice to improve biliary drainage [3,4]. SEMSs have longer patency times than those of plastic biliary stents. However, tumor ingrowth through the stent mesh, tumor overgrowth at the proximal or distal ends of the stent, compaction of biliary sludge or food, or mucosal hyperplasia due to stent-induced chronic inflammation can obstruct a SEMS [5,6].

Due to the fact that SEMSs have a mechanical drainage function only, without any antitumor effect, the drug-eluting stent (DES) was developed to suppress tumor ingrowth and increase stent patency time via chemical changes in the membrane [7,8,9,10,11,12,13]. Paclitaxel (hydrophobic) [7,8,9,10,11,12] and gemcitabine (2′, 2′-difluorodeoxycytidine; hydrophilic) [13] are the antitumor agents most frequently used in DESs. We found that local delivery of paclitaxel using a paclitaxel-eluting membrane inhibited growth of pancreatic/cholangiocarcinoma tumors in nude mice by suppressing angiogenesis via the mammalian target of rapamycin and induction of the apoptosis signaling pathway [14]. Gemcitabine is more effective at inhibiting pancreatic and biliary tumor cell growth compared with other chemotherapeutic drugs (e.g., paclitaxel) [15,16]. Hence, in this study, we examined the signal transduction pathways underlying the antiangiogenic and antiproliferative effects of a gemcitabine-eluting membrane (GEM) implanted in pancreatic/cholangiocarcinoma tumor-bearing nude mice.

We found that in addition to cell cycle regulatory effects, the GEM inhibited tumor growth and angiogenesis by suppressing the epidermal growth factor receptor (EGFR) signaling pathway. The GEM also inhibited tumor metastasis by suppressing the epithelial–mesenchymal transition (EMT), as well as angiogenesis and metastasis. The results indicated that GEM is a clinically relevant drug-eluting stent that suppresses cell proliferation/angiogenesis and metastasis of pancreatic/biliary tumors. Use of a GEM is a novel potential therapeutic strategy for inoperable malignant biliary obstructions.

## 2. Results

### 2.1. The GEM Inhibits the Growth of Pancreatic/Biliary Xenograft Tumors in Nude Mice

To determine whether gemcitabine suppresses pancreatic/biliary tumor growth, GEMs were fabricated for use in xenograft tumor models (Appendix A). After induction of tumor formation (60–80 mm^3^) by injecting SCK human cholangiocarcinoma cells or PANC-1 human pancreatic cancer cells, xenograft nude mice were implanted with a vehicle-eluting membrane (control) or GEM for 7 days (Appendix A). The GEM markedly reduced the growth of both PANC-1 and SCK tumor cells (Figure 1A). Consistent with this result, GEM implantation significantly reduced the volumes of the PANC-1 and SCK tumors in a dose-dependent manner at day 3, 5, and 7 (Figure 1B,D). Though volumes of PANC-1 and SCK tumors were significantly decreased in a dose-dependent manner, only 2% GEM implantation significantly decreased the wet weights of PANC-1 and SCK tumors at day 7 compared with the control group (Figure 1C,E). Histological analysis has revealed that GEM implantation restored cellular structure and organization compared with the control group (Appendix A). These results suggest that GEM implantation inhibits the growth of xenograft tumors.

### 2.2. GEM Implantation Inhibits EGFR Signaling and the Cell Cycle in Xenograft Tumors

Activation of the EGFR signaling pathway in cancer cells is associated with increased tumor growth, angiogenesis, and cell proliferation [17,18,19]. To identify the molecular mechanism by which GEM inhibits tumor growth, we investigated the effects of the GEM implant on EGFR phosphorylation and total protein levels. The phosphorylation and total protein levels of EGFR were reduced in the lysates of tumors implanted with the GEM compared with the control (Figure 2A,B). Activated oncogenic KRAS engages the PI3K/PDK1/AKT pathway to drive cancer initiation, progression, and maintenance [20]. KRAS mutation occurs in cancers such as pancreatic cancer, cholangiocarcinoma, colorectal cancer, and lung cancer [20]. The RAS protein level was markedly reduced in GEM-implanted tumor lysates (Figure 2A,B). We further investigated downstream proteins of EGFR/KRAS signaling in the GEM-implanted tumors. The GEM-implanted tumors exhibited decreased levels of PI3K, phospho-PDK, PDK, phospho-AKT, AKT, phospho-GSK3β, and GSK3β (Figure 2A,B). Due to the fact that cell cycle progression is tightly regulated by the expression levels of cyclins, we hypothesized that cyclin D1 and cyclin B expression would be reduced by GEM implantation. Indeed, cyclin D1 and cyclin B protein levels were markedly reduced in GEM-implanted tumor lysates (Figure 2C). Downregulation of cyclins was likely responsible for the arrest of mitosis at the G1/S transition and the subsequent inhibition of tumor growth. We next examined the levels of the modulators required to arrest cell cycle, such as including p53 and p21 in GEM-implanted tumors. GEM implantation resulted in dose-dependent increases in the protein levels of p53 and p21 (Figure 2C). The protein levels of cyclin D1 and cyclin B, but not p-ERK, decreased in a dose-dependent manner in the lysates of GEM-implanted SCK tumors (Figure 2D). Altogether, these results indicate that the GEM suppressed the EGFR signaling pathway and arrested the cell cycle to inhibit tumor growth.

### 2.3. GEM Implantation Induces Ubiquitination and Degradation of EGFR

To investigate whether the GEM decreased the EGFR protein level, we examined its effect on EGFR expression at the transcriptional level. The GEM did not affect the endogenous mRNA levels of EGFR as well as Ras (Figure 3A,B). Thus, the GEM decreases EGFR expression at the protein but not the mRNA level. The E3 ubiquitin ligase c-CBL is required for ubiquitin-dependent degradation of EGFR [21]. We found there was a tendency that the c-CBL protein levels were increased in high-dose GEM-implanted SCK and PANC-1 tumors (Figure 3C). Also, the GEM downregulated the EGFR protein level via ubiquitination in a dose-dependent manner in SCK and PANC-1 tumor lysates (Figure 3D). These results suggest that the GEM induces ubiquitination of EGFR, resulting in a degradation-mediated reduction in the EGFR protein level.

### 2.4. The GEM Reduces CD31 and Vascular Endothelial Growth Factor Receptor-2 (VEGFR2) Expression in Xenograft Tumors

We evaluated the mechanism of the antiangiogenic effect of the GEM because the gross findings indicated that tumor vascularization decreased with decreasing tumor size. Immunofluorescence staining of angiogenic markers in SCK tumor tissues was performed to determine whether the GEM reduced tumor growth by inhibiting angiogenesis. Compared with the control tumors, GEM implantation resulted in a significant decrease in staining of CD31 and VEGFR2 (Figure 4A,B). This indicated that the GEM inhibited tumor growth by suppressing tumor angiogenesis. The relative fluorescence intensities of CD31 and VEGFR2 are presented in Figure 4C,D. To evaluate the signal transduction pathways by which angiogenesis was inhibited in the GEM-implanted tumors, Western blot analysis was performed to measure the VEGF, VEGFR2, CD31, and HIF-1α protein levels in tumors from control and GEM-treated nude mice. The VEGF, VEGFR2, CD31, and HIF-1α protein levels decreased in a dose-dependent manner in the lysates of GEM-implanted PANC-1 and SCK tumors (Figure 4E). To evaluate drug penetration into the tumor tissues, we analyzed SCK tumor tissue bottom (proximal to the GEM) and top (distal to the GEM) cross-sections. VEGF, CD31, and HIF-1α expression decreased in a dose-dependent manner in the lysates (bottom and top) of GEM-implanted SCK tumors (Figure 4F). Taken together, these results indicate that local delivery of gemcitabine increases its antiangiogenic activity at tumor sites and potentiates its therapeutic efficacy.

### 2.5. GEM Implantation Suppresses the EMT-like Features in Xenograft Tumors

Induction of the EMT in epithelial tumor cells enhances migration and invasion [22,23]. To investigate whether the GEM inhibited invasiveness by repressing the EMT, we examined its effects on migration and invasion in tumor cells. We found that GEM implantation significantly decreased tumor cell migration and invasion in PANC-1 and SCK tumors compared with the control groups (Figure 5A,B). Acquisition of mesenchymal markers, such as N-cadherin, snail, and vimentin, in carcinoma cells promotes the EMT with concomitant loss of epithelial E-cadherin, a major constituent of the adherens junctions [23,24]. We performed Western blot analysis to examine the effect of the GEM on the protein levels of E-cadherin, N-cadherin, snail, and vimentin in PANC-1 and SCK cells. As the gemcitabine dose was increased, the level of E-cadherin increased in a dose-dependent manner, whereas the levels of N-cadherin, snail, and vimentin decreased (Figure 5C,D). Immunofluorescence staining in the SCK xenograft tumor tissue indicated significantly increased E-cadherin and decreased N-cadherin, snail, and vimentin protein levels after GEM implantation compared with the control tumors (Figure 5E). Consistently, αSMA mRNA levels were remarkably reduced by GEM (Appendix A). Together, these results suggest that the GEM inhibits tumor metastasis by suppressing the EMT-like features.

## 3. Discussion

Gemcitabine is a nucleoside analog that exhibits antitumor activity and is widely used as the standard treatment for locally advanced and metastatic pancreatic carcinoma. Gemcitabine exhibits cell phase specificity, primarily killing cells in the S phase and blocking the progression of cells through the cell cycle (i.e., from the G_1_ to S phases) [25]. Gemcitabine has antiproliferative and antiangiogenic activities and gemcitabine-based chemotherapy is accepted as the only effective systemic therapy for pancreatic/biliary cancer [26]. However, the clinical application of gemcitabine for cancer treatment is limited by its poor availability when given systemically. A phase III randomized trial was performed to examine the efficacy of gemcitabine combined with oxaliplatin, with or without erlotinib, in patients with biliary tract cancer [27]. Although a significant increase in the response rate was observed, there was no improvement in survival time with the combination treatment [27]. Therefore, development of an alternative gemcitabine delivery system that increases its availability at tumor sites and maximizes its therapeutic efficacy while minimizing systemic side effects is ongoing [28,29]. Moon et al. found that controlled release of gemcitabine from polyamide/polytetrafluoroethylene (PA/PTFE)-covered DESs increases the usefulness of DESs for the treatment of malignant gastrointestinal cancer and cancer-related stenosis [30]. However, the antiproliferative and antiangiogenic effects of the GEM for pancreatic/biliary cancer and the underlying molecular mechanisms were unclear. We found that the GEM is a local drug delivery (LDD) device capable of delivering gemcitabine to tumors and inhibiting tumor cell growth, angiogenesis, and metastasis by suppressing EGFR signaling pathways.

EGFR is a transmembrane receptor tyrosine kinase that promotes malignant behaviors, including cell cycle progression, proliferation, angiogenesis, and cell survival [31,32]. EGFR is generally overexpressed or mutated in many solid malignancies and is associated with a poor clinical prognosis. As a result, EGFR is a therapeutic target for cancer [33,34]. We found that the GEM significantly decreased the phosphorylation and protein levels of EGFR in a dose-dependent manner in PANC-1 and SCK cells, with no change in the endogenous mRNA level of EGFR. This result indicated that the GEM directly targets EGFR to reduce its endogenous protein level. The c-CBL E3 ubiquitin ligase is responsible for the ubiquitination and subsequent internalization of EGFR into early endosomes for degradation [35,36]. We found that the GEM increased the c-CBL protein level in a dose-dependent manner, resulting in ubiquitination and degradation of EGFR in SCK and PANC-1 cells. These results suggest that local delivery of gemcitabine via a GEM implant induces ubiquitination and degradation of EGFR, leading to inhibition of the EGFR-mediated signaling pathway.

The Ras/PI3K/PTEN/AKT/mTOR pathway mediates EGFR downstream signaling, playing a key role in the regulation of proteins critical for protein translation [37]. Activation of EGFR and the Ras/PI3K/PTEN/AKT/mTOR signaling pathway also triggers angiogenesis by upregulating HIF-1α and inducing the expression of VEGF [38]. GEM-implanted tumors exhibited reduced protein levels of EGFR, phospho-EGFR, Ras, PI3K, phospho-PDK, PDK, phospho-AKT, AKT, phospho-mTOR, phospho-GSK3β, and GSK3β. This indicated that local delivery of gemcitabine via a GEM implant inhibits tumor growth by suppressing EGFR signaling in xenograft tumors.

Tumor progression requires angiogenesis, which is generally induced in response to hypoxia via the angiogenic switch process [39]. VEGF and its signal-transducing tyrosine kinase receptor, VEGFR-2, are major mediators of angiogenesis [39]. VEGF/VEGFR2 signal transduction activates downstream signaling molecules responsible for endothelial cell migration, proliferation, and survival under hypoxic conditions [40]. We found that in both the pancreatic and cholangiocarcinoma models, GEM implantation reduced tumor growth by affecting vascularization by reducing VEGF, VEGFR2, CD31, and HIF-1α protein levels. Therefore, the GEM inhibited the growth of pancreatic cancer and cholangiocarcinoma cells by suppressing angiogenesis.

Cell cycle progression is directly associated with angiogenesis and tumor growth. Modulation of the expression levels and activities of cell cycle proteins, such as cyclins and CDK inhibitors, is an important mechanism of cell proliferation [41,42]. Cyclin B1 is a regulatory subunit of the mitosis- promoting factor, and its proper regulation is essential for the initiation of mitosis [43]. We found that GEM implantation reduced tumor growth by downregulating cell cycle modulators promoting cell proliferation, particularly cyclin D1 and cyclin B, and upregulating the cell cycle modulators arresting cell cycle, such as p53 and p21. Therefore, the GEM implant suppresses tumor growth by directly modulating the protein levels of cell cycle regulators.

In the EMT, polarized epithelial cells lose their polarity and cell–cell adhesion and gain invasive properties, resulting in adoption of a mesenchymal cell phenotype [44,45]. Invasive tumor cells generally show evidence of the EMT, such as reduced E-cadherin and increased N-cadherin, snail, and vimentin expression [46]. Wang et al. found that gemcitabine-resistant pancreatic cancer cells acquire an HIF1α-mediated EMT phenotype with increased migration and invasion properties [47]. Therefore, targeting HIF-1α could be an effective strategy for the treatment of pancreatic cancer. We found that the GEM decreased the HIF-1α protein level in a dose-dependent manner in SCK and PANC-1 tumors. This resulted in decreased tumor cell migration and invasion. Consistent with this result, GEM implantation resulted in a significant increase in the E-cadherin level and decreases in the N-cadherin, snail, and vimentin levels in the tumor cells. Therefore, GEM implantation inhibits tumor metastasis by reducing the rate of HIF1α-mediated EMT.

## 4. Materials and Methods

### 4.1. Cell Lines and Antibodies

SCK human cholangiocarcinoma cells and PANC-1 human pancreatic cancer cells were cultured in Roswell Park Memorial Institute (RPMI)-1640 medium. The SCK cells were procured from Dr. Dae-Ghon Kim of Chonbuk National University Medical School and Hospital (Jeonju, South Korea). The PANC-1 cells were purchased from the American Type Culture Collection (Manassas, VA, USA). The cells were maintained in a humidified incubator at 37 °C in an atmosphere containing 5% CO_2_. Antibodies against EGFR, phospho-EGFR, PI3K, PDK, phospho-PDK, AKT, phospho-AKT, GSK3β, phospho-GSK3β, ERK, phospho-ERK, cyclin B1, vimentin, and GAPDH were obtained from Cell Signaling Technology (Danvers, MA, USA). CD-31 and VEGF were purchased from Abcam (Cambridge, MA, USA). HIF-1α, VEGFR2/FLK-1, p53, p21, cyclin D1, E-cadherin, N-cadherin, snail, c-CBL, and RAS were obtained from Santa Cruz Biotechnology (Santa Cruz, CA, USA).

### 4.2. Mouse Xenograft Experiment

Female athymic nude mice (6–8 weeks old) were purchased from Orient Bio (Kyunggido, South Korea) and used for subcutaneous xenograft experiments. To establish the tumor xenograft model, 2 × 10^6^ of PANC-1 and SCK cells were suspended in 200 µL RPMI-1640 medium. The cells were injected into the subcutaneous space under the dorsal skin. Tumor size was measured every other day using calipers, and tumor volume was calculated using the formula: 0.5 × tumor length × tumor width [2]. The body weights of the mice were measured every other day after GEM implantation (Appendix A). When the tumor volume reached 100 mm^3^, the mice were anesthetized with isoflurane (1.5%–3% in oxygen, Troikaa Pharmaceuticals Ltd., Ahmedabad, Gujarat, India). The GEMs were then surgically implanted beneath the tumors. All animal studies were performed in compliance with the policies of the Animal Care and Use Committee of the Korean Research Institute of Bioscience and Biotechnology (2015-0397, approval date 21 December 2015).

### 4.3. Preparation of the GEM

Briefly, 400 mg polymer (polyurethane) was dissolved in 10 mL tetrahydrofuran (THF). Pluronic F-127 (40 mg) and ~0.1–5 wt.% gemcitabine (Dong-A Pharmaceutical co., Seoul, South Korea) were dissolved in distilled water, added to THF, and mixed by vortexing and sonication. The mixture (200 µL) was poured into a dish-shaped Teflon mold. After air drying, the dish-shaped GEM was carefully peeled off the Teflon mold. After serial dilution of the membrane in THF, the amount of gemcitabine in the membrane was estimated using an ultraviolet spectrophotometer (UV-1601, Shimazu Co., Kyoto, Japan).

### 4.4. Tumor Sample Immunofluorescence

The tumors were immersed in Optimum Cutting Temperature™ compound (Leica Biosystems, Richmond, CA, USA), frozen in liquid nitrogen, and sectioned at a 5 µm thickness. The sections were permeabilized, blocked for 60 min with 10% goat antiserum and 0.25% Triton X-100 in phosphate-buffered saline to prevent nonspecific binding, and subsequently incubated with an anti-CD31 (1:200), anti-VEGFR2 (1:200), anti-E-cadherin (1:100), anti-N-cadherin (1:100), anti-snail (1:100), or anti-vimentin (1:100) antibody. Next, the sections were washed and incubated with the corresponding Alexa-488-conjugated secondary antibodies (Invitrogen, Carlsbad, CA, USA). Nuclei were stained with 4′,6-diamidino-2-phenylindole (DAPI) (Invitrogen). Fluorescence images were acquired using a confocal microscope (Carl Zeiss, Oberkochen, Germany). The fluorescent intensity was acquired by image analysis of Zen software provided by Carl Zeiss.

### 4.5. Western Blot Analysis

The tumors were minced coarsely and homogenized in lysis buffer containing 100 mM tris (pH 7.4), 150 mM NaCl, 1% Triton X-100, 15% glycerol, 1 mM phenylmethanesulfonyl fluoride, phosphatase inhibitor mixtures 2 and 3 (Sigma, St. Louis, MO, USA), and a protease inhibitor mixture (Sigma). The homogenates were centrifuged at 14,000 rpm and 4 °C, and the supernatants were used for analysis. Protein concentration was determined by bicinchoninic acid assay using bovine serum albumin as the standard. Tumor lysates containing 60 µg protein were resolved by sodium dodecyl sulfate–polyacrylamide gel electrophoresis and transferred to nitrocellulose membranes (Bio-Rad, Hercules, CA, USA). The membranes were incubated for 2 h with antibodies against the target proteins. After washing, the membranes were incubated with the corresponding secondary antibody, and the protein bands were detected using enhanced chemiluminescence reagents (West Pico PLUS, ThermoFisher, Waltham, MA, USA). Band intensities have been quantified by Image J (Appendix A).

### 4.6. Migration Assay

The chemotactic motilities of SCK and PANC-1 cells were evaluated by Transwell migration assay [48]. Transwells (Corning Costar, Cambridge, MA, USA) with polycarbonate filters 6.5 mm in diameter (8 µm pore size) were used for the assays. Briefly, the lower surface of the filter was coated with 10 µg gelatin. The SCK and PANC-1 cells were trypsinized and suspended in serum-free RPMI medium at a final concentration of 1 × 10^6^/mL. Fresh RPMI medium (10% fetal bovine serum) containing gemcitabine was placed in each lower well, and 100 µL of the cell suspension was loaded into each upper well. The chamber was incubated at 37 °C for 4 h, and the cells were fixed and stained with hematoxylin and eosin. Nonmigrating cells on the upper surface of the filter were removed by wiping with a cotton swab. Chemotaxis was quantified by counting the cells that migrated to the lower side of the filter; the cells were visualized using an inverted microscope at ×100 magnification.

### 4.7. Invasion Assay

Cell invasion assays were performed using Transwells similarly to the cell migration assay, except that the membrane was coated with Matrigel basement membrane matrix (BD Biosciences catalog number 354230, CA).

### 4.8. RNA Extraction and Reverse Transcription–Polymerase Chain Reaction

Total RNAs from transplanted tumors were isolated using the TRIzol Reagent Kit (Invitrogen, Carlsbad, CA, USA). To prepare samples for reverse transcription–polymerase chain reaction, 5 µg mRNA were converted to cDNA using 200 U reverse transcriptase and 500 ng oligo-dT18 primer in buffer comprising 50 mM tris-HCl (pH 8.3), 75 mM KCl, 3 mM MgCl_2_, 10 mM diothiothreitol, and 1 mM dNTPs at 42 °C for 1 h. The reaction was stopped by heating at 70 °C for 15 min; 1 µL of the cDNA mixture was used for enzymatic amplification. PCR was performed using AccuPower PCR Premix (Bioneer, Daejeon, Korea) according to the manufacturer’s protocol. The PCR parameters consisted of denaturation at 94 °C for 1 min, annealing at 60 °C for 30 s, and extension at 72 °C for 2 min for 30 cycles. Band density was determined by densitometry using a computer-assisted image analyzer. The primers used were 5′-CTTCTTGCAGCGATACAGCTC-3′ (sense) and 5′-ATGCTCCAATAAATTCACTGC-3′ (antisense) for EGFR and 5′-TTAGCTGTGCTCGCGCTACTCTCTC-3′ (sense) and 5′-GTCGGATTGATGAAACCCAGACACA-3′ (antisense) for actin as the internal control.

### 4.9. Statistical Analysis

Results are presented as means ± standard deviation. Statistical comparisons were performed by post hoc group comparison in ANOVA with SAS version 9.4. A difference was considered statistically significant if the *p*-value was <0.05. Each experiment was performed in at least triplicate.

## 5. Conclusions

In conclusion, local delivery of gemcitabine via implantation of a GEM inhibited the growth of pancreatic cancer and cholangiocarcinoma cells in a mouse xenograft model. GEM implantation reduced the protein level of EGFR by promoting its c-CBL-mediated ubiquitination and degradation, thus inhibiting its downstream effects, such as tumor cell growth, angiogenesis, and metastasis (Figure 6). Understanding the molecular mechanisms of action of gemcitabine-eluting stents will improve the treatment of patients with malignant biliary obstruction.

## Figures and Tables

**Figure 1 ijms-21-01605-f001:**
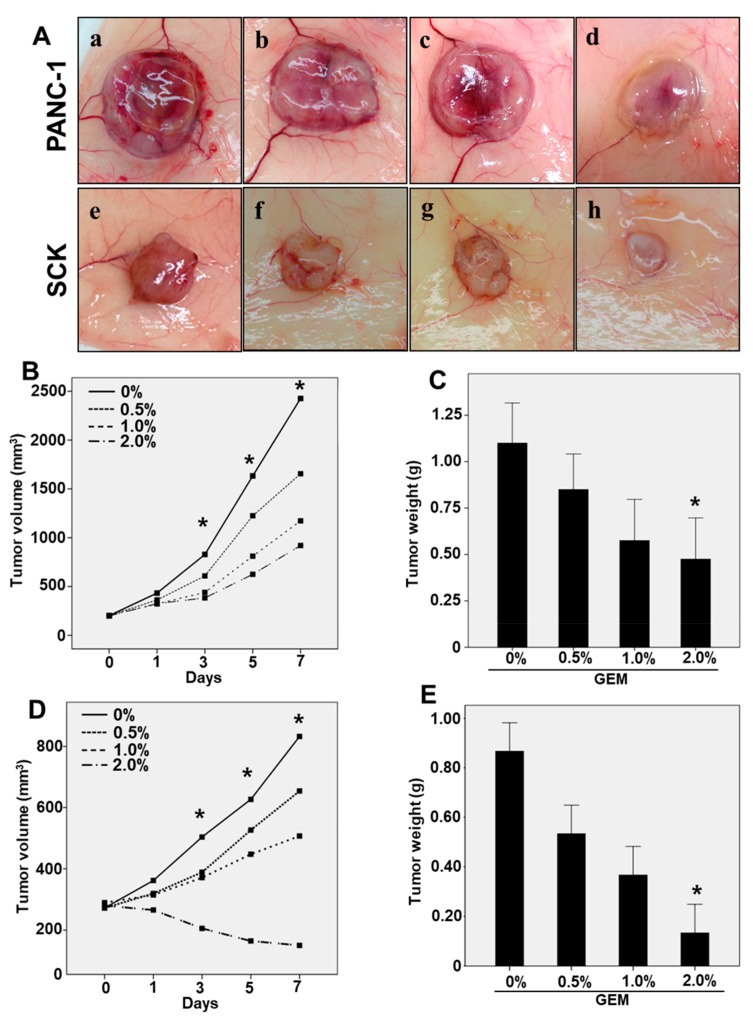
Use of a gemcitabine-eluting membrane (GEM) reduced tumor growth in a mouse xenograft model. (**A**) Representative photographs of the gross morphology of PANC-1 and SCK tumors from nude mice implanted with the control (0% Gemcitabine), (**a**,**e**); Gemcitabine 0.5%, (**b**,**f**); Gemcitabine 1%, (**c,g**); Gemcitabine 2%, (**d**,**h**); (**B**) The mean of PANC-1 tumor volumes implanted with 0%, 0.5%, 1%, or 2% GEM for 7 days. The volumes of PANC-1 tumors were statistically decreased at day 3, 5, and 7. (**C**) The wet weights of PANC-1 tumors implanted with 0%, 0.5%, 1%, and 2% GEM at 7 days. The weights of PANC-1 tumors implanted with 2% GEM were significantly decreased at day 7 compared to control group. (**D**) The mean of SCK tumor volumes implanted with 0%, 0.5%, 1%, or 2% GEM for 7 days. The volumes of SCK tumors were statistically decreased at day 3, 5, and 7. (**E**) The wet weights of SCK tumors implanted with 0%, 0.5%, 1%, and 2% GEM at 7 days. The weights of SCK tumors implanted with 2% GEM were statistically decreased at day 7 compared to control group. Overall *p* value was evaluated by ANOVA * *p* < 0.05.

**Figure 2 ijms-21-01605-f002:**
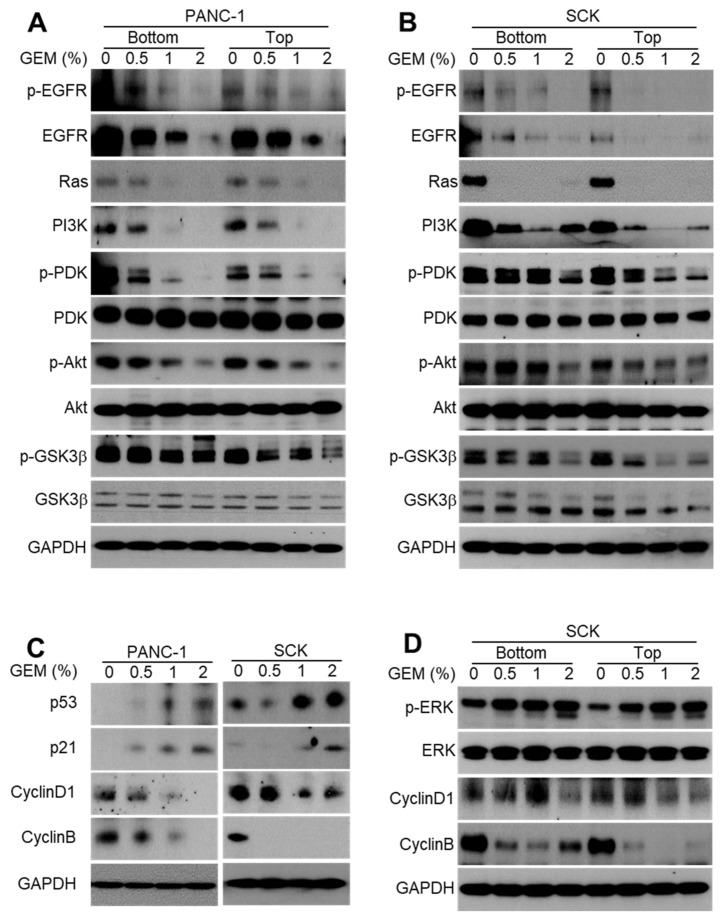
Gemcitabine-eluting membrane (GEM) implantation inhibits EGFR signaling in xenograft tumors. (**A**,**B**) Western blot analysis of phospho-EGFR, EGFR, Ras, PI3K, phospho-PDK, PDK, phospho-AKT, AKT, phospho-GSK3β, and GSK3β at the bottom and top of the indicated tumors from the GEM-implanted and control groups. (**C**) p53, p21, cyclin D1, and cyclin B expression was evaluated by Western blot analysis in SCK and PANC-1 tumors after implantation of the GEM containing the indicated concentrations of gemcitabine. (**D**) Proteins derived from tumor lysates obtained from the bottom and top of SCK tumors were subjected to Western blot analysis using the indicated antibodies.

**Figure 3 ijms-21-01605-f003:**
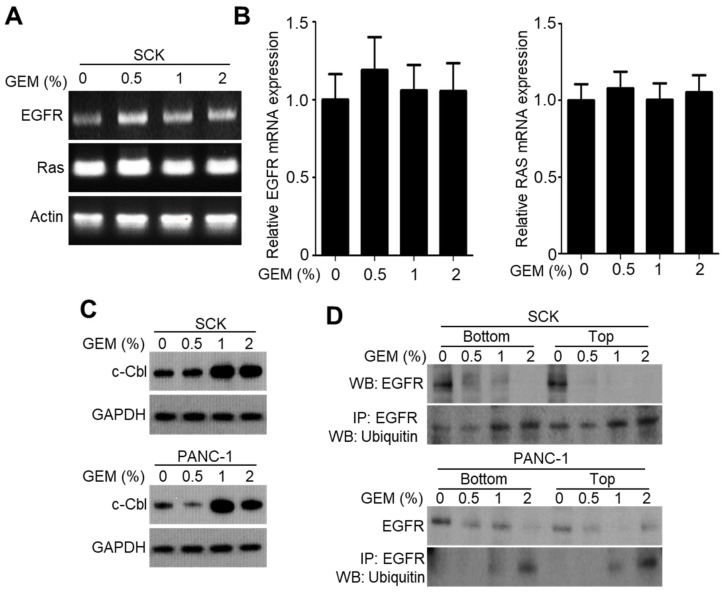
Gemcitabine-eluting membrane (GEM) implantation reduces EGFR expression by promoting c-CBL-mediated EGFR degradation. (**A**,**B**) Reverse transcription–polymerase chain reaction was performed to evaluate EGFR expression in SCK tumor lysates after implantation with GEM containing the indicated concentrations of gemcitabine. (**C**) Western blot analysis of c-CBL in SCK and PANC-1 tumor lysates after GEM implantation containing the indicated concentrations of gemcitabine. (**D**) The level of ubiquitinated EGFR was determined by immunoprecipitation using an anti-EGFR antibody.

**Figure 4 ijms-21-01605-f004:**
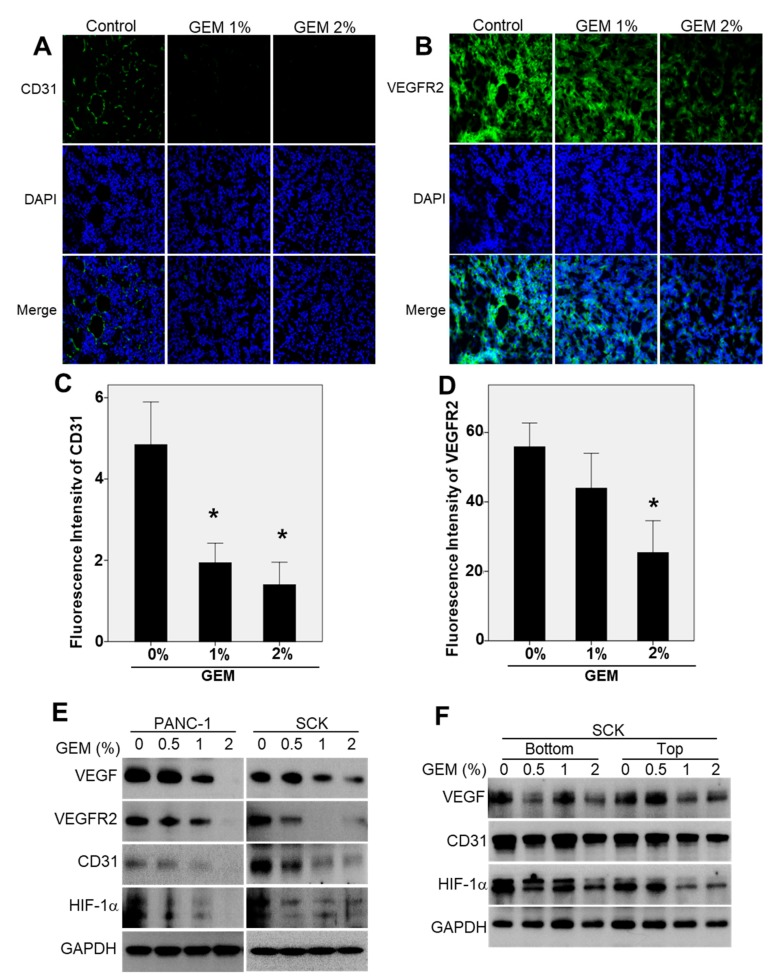
Gemcitabine-eluting membrane (GEM) implantation attenuates angiogenesis in xenograft tumors. Representative results of immunofluorescence analyses of (**A**) CD31 and (**B**) VEGFR2 in SCK tumors implanted with the GEM. Positive protein staining is indicated by green, and nuclei are stained blue with DAPI. Original magnification, ×20. *n* = 7 fields per group. (**C**,**D**) The relative fluorescence intensities of CD31 and VEGFR2. (**E**) Western blot analysis of VEGF, VEGFR2, CD31, and HIF-1α expression in SCK and PANC-1 tumors after implantation with the GEM containing the indicated concentrations of gemcitabine. (**F**) Western blot analysis of VEGF, CD31, and HIF-1α at the bottom (proximal to the GEM) and top (distal to the GEM) of SCK tumor tissues after GEM implantation. Overall *p* value was evaluated by ANOVA. * *p* < 0.05.

**Figure 5 ijms-21-01605-f005:**
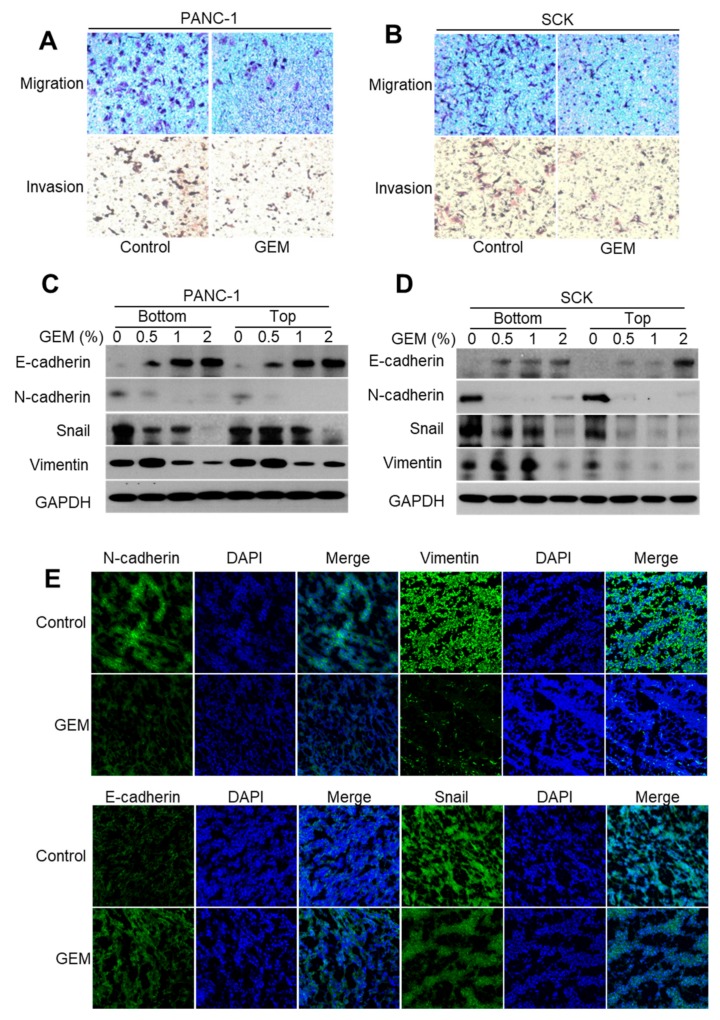
Gemcitabine-eluting membrane (GEM) implantation suppresses the epithelial–mesenchymal transition (EMT) in xenograft tumors. (**A**,**B**) Transwell migration and invasion assays of SCK and PANC-1 cells. (**C**,**D**) Western blot analysis of E-cadherin, N-cadherin, snail, and vimentin expression at the bottom and top of SCK and PANC-1 tumor tissues after implantation with the GEM containing the indicated concentrations of gemcitabine. (**E**) Results of representative immunofluorescence analyses of E-cadherin, N-cadherin, snail, and vimentin in tissues from SCK tumors implanted with the GEM or control membrane.

**Figure 6 ijms-21-01605-f006:**
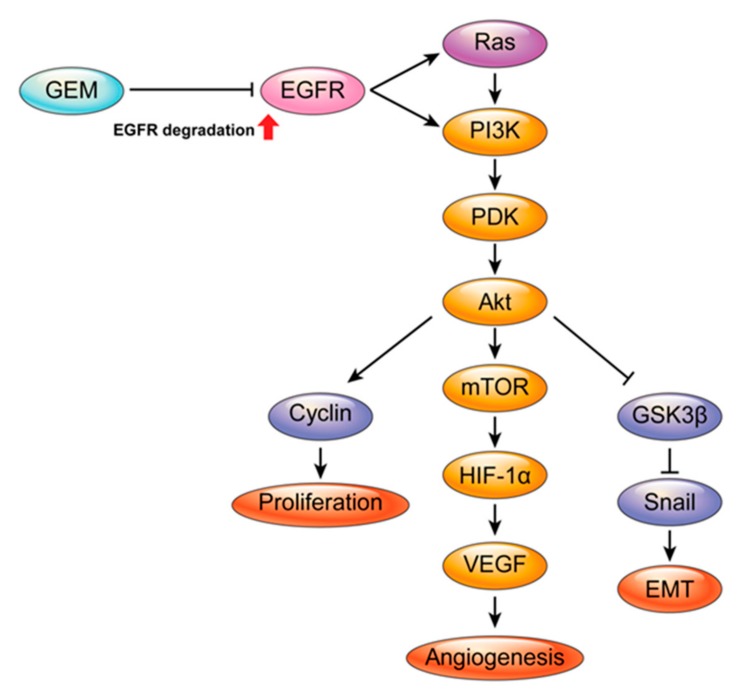
Molecular mechanisms of a gemcitabine-eluting membrane (GEM). Local delivery of gemcitabine via implantation of a GEM induces degradation of epidermal growth factor receptor (EGFR), suppression of EGFR downstream signaling pathways (e.g., cell proliferation, angiogenesis, and the EMT), and reduction of tumor growth.

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
