# Peer review of "Local Delivery of Gemcitabine Inhibits Pancreatic and Cholangiocarcinoma Tumor Growth by Promoting Epidermal Growth Factor Receptor Degradation"

_ijms, 2020, doi:10.3390/ijms21051605_

Round 1

Reviewer 1 Report

In this manuscript, Jang and colleagues evaluated the effect of local treatment with Gemcitabine on the development of 2 primary epithelial neoplasms, cholangiocarcinoma and pancreatic carcinoma. Authors analyzed both in vitro and in vivo the influence gemcitabine-eluting membranes (GEMs) on tumoral overgrowth following xenograft of human cancer cells in athymic nude mice. Moreover, Authors evaluated in vitro the effect on cell proliferation, and in vivo on neoangiogenesis and EMT and the molecular pathways mediating those features, following treatment with different concentration of gemcitabine or GEMs, using 2 different human established tumoral cell lines, SCK (cholangiocarcinoma) and PANC-1 (pancreatic tumor). The most important results reached by the Authors are that a.) treatment with different concentration of gemcitabine inhibit proliferation on both neoplastic cell lines in vitro; in vivo, the use of GEM inhibit b.) the neoangiogenesis and VEGF expression respect to controls, and c.) modulates the expression of several EMT-related markers.

The paper is clear and well written and data are potentially of interest but some flows significantly dampen the enthusiasm for this yet interesting paper.

Major concerns:

Hematoxylin and eosin stains of the explanted tumor masses must be added to better evaluate the morphology of the tumor originating from the xenografts. Authors showed only (Figures 4 and 5) in vivo stainings for xenotransplants deriving from SCK cells; what about PANC-1? Reviewer suggests to perform stainings also for VEGF and Hif1α on tumor masses to reinforce western blot data. Is not clear to the Reviewer, how Authors evaluated “intensity” of CD31 and of VEGFR2, please discuss; moreover, Reviewer things that the extent of positive vascular structures could be more interesting respect to fluorescence intensity for evaluating the neoangiogenesis and that the evaluation of the microvessel density (MVD) could be added. To state that epithelial cells undergo a proper EMT, Authors have to evaluate the expression of aSMA; if negative on epithelial cells, the Reviewer suggest to refer to this feature as EMT-like features and not as EMT.

Minor concerns:

Some edits on font size must be done (page 1 line 40). Figure S1 and S2 must be transferred in an appropriate on-line section and not in the text (since both are marked as “supplementary”) (page 2 and 3).

Reviewer 2 Report

This work investigated the application of gemcitabine-eluting membranes (GEMs) in mouse models with both pancreatic cancer and cholangiocarcinoma cell subcutaneous xenografts. The GEMs possess dual functions of mechanically improving biliary drainage, as well as direct and sustained delivery of a chemotherapeutic agent. The authors demonstrate GEMs ability to reduce tumor volume and weight in both cancer xenograft models, as well as characterize the mechanism of this via an array of different important proteins in several pathways that are heavily investigated in cancer biology. Their main techniques involve xenografts, gross anatomical observation, Western blotting, and immunofluorescence.

  Overall I would say accept with minor changes.     The following are my concerns. Some of them are major, though most of them are minor:   Snail should have the first letter capitalized to conform with standard format in the literature, even though it is clear in context that they are not referring to the gastropod.   Figure S1 and S2 are inserted directly in the main manuscript, but called supplemental and also submitted separately in the supplemental figures. Are these intended to be supplemental figures or not?   Line 68 a claim is made about reduction in vascularization, though no direct evidence specific to vascularization is offered. Remove this portion of the claim, or undergo some direct or indirect measurement, e.g. tumor blood volume, MVD, etc. Gross anatomical examination is insufficient for a direct claim.   Fig 1 the control (F-127) membrane from gross morphology appears to be significantly different in size than the 0% GEM implant, though no quantification is given. Also, 2% GEM implant data is shown, but no gross morphology image. In general, no quantitative comparison seems to be done between F-127 and 0% GEM, so it is unclear why they claim it is a control. It could be an important control, but is not used effectively as such.   Line 86 and 87 for Fig 1 legend: it states that 1% or 2% GEM are significant, though the figure itself has * also by 0.5% to indicate statistical significance.    Fig 2 likely the entire difference in p-EGFR is due to difference in total EGFR, since the pattern is identical.     Several of the WBs have very saturated bands, as well as bands that have merged from other lanes. Examples include, but are not limited to: Fig 2A p-EGFR and Fig 4E HIF-1a. At least the authors did not attempt "quantification" with WB, so it is probably ok for qualitative comparisons.   Why was RT-PCR only done for EGFR and not others, e.g. RAS? Why was it only done for SCK and not PANC-1 (3A & 3B)? This should be investigated more precisely, since line 126 directly claims that reduction occurs at the protein and not mRNA level.   Why was p-mTOR measured by WB but not also total mTOR, as was done for other proteins, e.g. p-PDK vs PDK?   Fig 3B the axis label should read EGFR instead of ERFR.   The authors claim a dose-dependent increase in c-CBL is shown in Fig 3C (line 128-129); however, it appears that at 0.5% it is equal to (SCK) or less than (PANC-1) the 0%, and 1% is greater than either 2% or 5%. This is clearly not in direct agreement with their statement.   Some blots are either too low resolution or not exposed long enough. For example, Figure 3D, particularly for PANC-1   Line 288 the body weight was supposedly measured every other day, but it is never mentioned or shown. Perhaps a supplemental figure with the body weight and a mention in the discussion would be helpful. Otherwise, no examination whatsoever is given to show that the GEM implants are not adversely affecting the mice. In my opinion, even simply showing body weight stability is not sufficient evidence for the safety of the chemotherapy-infused implants. Overall, I would say this represents a significant weakness of the study. It appears that no real in vitro or in vivo experiment was done to assess toxicity or other adverse impact from the GEM implants.   Section 4.7 was complete or growth factor reduced matrigel utilized? Since the mechanism for GEM involves EGF, this could be an important consideration. It is perhaps worth testing both.   Section 4.9 in many cases, it is not the most appropriate to use a Student's t-test, particularly for the manner they have measured multiple treatments and multiple days with tumor volume (e.g. Fig 1B and 1D). For this, an ANOVA would be more appropriate, if desired with post-hoc analysis incorporating correction for multiple statistical tests on the same data. Or, if one insists on multiple t-tests, as with e.g. tumor weight Fig 1C & 1E, then the Holm method with Šídák modification could be used. This is all easily accomplished with a few clicks using GraphPad. This is important because error propagates from using multiple tests, which is not accounted for, and the assumption of type II statistical error set as a=0.05 is not correct, unless the above-mentioned or other modifications to the analysis are applied. 

Round 2

Reviewer 1 Report

Authors answered most of the requests and concerns raised by the Reviewer; paper is definitely improved.

Author Response

Thank you for raing the isses for the mauscript!

1) We modified the subheading 2.5 and the last sentence of the section following the reviewer's comments. All changes have been highlightened in the revised manscript.

2) Thank you for raising issue on figure 1 legend that we have missed. Since we already described our figure 1 data in 2.1 carefully, we just changed our figure 1 legend very briefly. All changes have been highlightened in the revised manuscript. We think that the audience would not be confused any longer with figure 1 legend any longer.

We really appreciate the reviewer's precious comments to improve our manscript.